# Cultural Influences on Family Mealtime Routines in Mexico: Focus Group Study with Mexican Mothers

**DOI:** 10.3390/children9071045

**Published:** 2022-07-14

**Authors:** Elizabeth Villegas, Amber J. Hammons, Angela R. Wiley, Barbara H. Fiese, Margarita Teran-Garcia

**Affiliations:** 1Department of Early Childhood, Child Trends, 7315 Wisconsin Ave, Ste 1200W, Bethesda, MD 20814, USA; 2Department of Child and Family Science, California State University Fresno, 5300 N Campus Dr. M/S FF12, Fresno, CA 93740, USA; ahammons@csufresno.edu; 3Department of Human Development and Family Studies, Auburn University, 210 Spidle Hall, Auburn, AL 36849, USA; arw0044@auburn.edu; 4Department of Human Development and Family Studies, Family Resiliency Center, University of Illinois at Urbana-Champaign, 904 W Nevada St., Urbana, IL 61801, USA; bhfiese@illinois.edu; 5Integrated Health Disparities Programs, University of Illinois Extension, 111 Mumford Hall, 1301 W. Gregory Dr., Urbana, IL 61801, USA; teranmd@illinois.edu; 6Division of Nutritional Sciences, Carle Illinois College of Medicine, Family Resiliency Center, University of Illinois at Urbana-Champaign, 2103 N. Dunlap Ave., Suite B, Champaign, IL 61820, USA

**Keywords:** mealtime routines, hispanic families, globalization, acculturation

## Abstract

Mexico’s obesity rates are alarming, and experts project drastic increases in the next thirty years. There is growing interest in understanding how remote acculturation and globalization processes influence health behaviors. The present study used focus group data from a central state, San Luis Potosí, in Mexico to explore mothers’ perspectives on factors that influence family mealtime routines. Thematic analysis was used to identify barriers and facilitators to healthy mealtime routines and diet within families. Twenty-one mothers participated in the focus groups; 76% were full-time stay at home mothers, the majority were low-income (65%), and fewer than half reached a high school education. Three major themes emerged: (1) mothers do most of the visible work involving mealtime routines; (2) family meals are different today, and globalization is a contributing factor; and (3) family mealtimes are shifting to weekend events. Empowering mothers to negotiate diet with family members to build healthy routines, navigate challenges due to changing environments, and set family boundaries around technology use during mealtimes should be taken into consideration when promoting healthy behaviors with Mexican families.

## 1. Introduction

The Organization of Economic Co-operation and Development (OECD) reported that more than 70% of adults are overweight (32% are obese) in Mexico, with one in three children overweight. Mexico is ranked third among all countries studied under the OECD [1]. This alarming percentage of overweight and obesity has spurred researchers to explore the role of cultural and geographical influences in this public health crisis. Cultural and geographical influences that may contribute to visible changes and reorganizations in the family structure and routines are of particular interest [2]. Key factors of globalization (e.g., technology, media, goods) have infiltrated various cultures through the introduction of new and different cultural practices [3]. One form of acculturation that can occur among nonmigrants is remote acculturation, defined as “the exposure of nonmigrants to remote cultures in which they have never lived” [4] (p. 157). It often occurs through indirect and/or intermittent cultural contact from modern trade, media, and technology and may play a similar role for families born and raised in Mexico who do not migrate [5]. Behavioral changes stemming from acculturation and the implications of cultural shifts can present barriers to a healthy lifestyle. Globalization is a major component with potential to affect diet and family mealtime routines, which in turn may affect parent and child weight status [6,7].

Cultural norms and practices include food and meal choices and how families share meals [8]. A systematic review of the traditional Mexican diet found that grains, tubers, legumes, and vegetables were most commonly consumed across studies; however, the food culture in Mexico continues to change, making it challenging to establish a robust definition of the traditional Mexican diet [9]. Historically, traditional Hispanic food patterns included fresh and local healthy foods [10], but families’ diets today include more processed foods and sugar-sweetened beverages [11]. Shared family meals are viewed as central for the celebration and unity of family, and as such, meals should be regularly occurring and involve as many family members as possible [12]. Gender role divisions have been prevalent in Hispanic culture, with mothers typically having authority in matters related to food and mealtimes [13]. Respect for parental authority has also been a traditional cultural norm, with children behaving as passive participants of mealtime routines, allowing mothers to cook and make the decisions involving food [14]. 

Several factors present challenges to these traditional patterns. Household dynamics, such as financial burdens, have also been associated with difficulty allocating time and energy to execute daily meals [15]. Furthermore, communication among Hispanic families may look different today in the face of increased distractions with T.V. viewing or phone usage during mealtimes [16]. These barriers can make it challenging to maintain adequate and positive communication during mealtimes, creating opportunities for unhealthy diet [17,18]. Research has begun to examine Hispanic immigrant parents’ perceptions of the family food environment in more recent years, but Mexican families living in Mexico are understudied. Understanding how mealtime routines are experienced in Mexican families is important in advancing the knowledge of the roles globalization and cultural factors play in maintaining family practices and their potential impact on family health. 

Apart from these micro-level family cultural shifts, western media may also play a role in the rising overweight and obesity rates among children and adolescents around the world [19]. Across the globe, exposure to junk food in advertising, entertainment, television, games, movies, and other streaming services has been related to increased children’s consumption of energy dense, low nutrition products [20,21]. Remote influences from the United States on food, nutrition, and family routines can have changing effects on families across the globe, especially Mexico, due to its proximity as a bordering country. Understanding how cultural shifts and globalization play a role in mothers’ perceptions of the family food environment may be helpful in designing interventions to improve healthier living, particularly as it relates to nutrition and healthy eating. This is especially important in Mexico where the rates of obesity are currently among the highest in the world. 

Exploring Mexican mothers’ perceptions of mealtime routines and the changing food environment may reveal familial and cultural factors that are important in understanding the rapid increase in obesity rates in Mexico. The ways in which families continually share meals, including discussion that takes place during meals, and the patterns that emerge from shopping, cooking, mealtime preparation, and mealtime rules, roles, and expectations all contribute to family mealtime routines [11], and are also critical in understanding how these family practices may influence family health. Therefore, the present study sought to explore (a) the division of family roles and how those roles influence mealtime routines; (b) mothers’ childhood mealtime routines and how these may influence the practice of current family mealtime routines; and (c) global influences and how new environmental factors may have shifted mealtime and cultural practices pertaining to mealtimes.

## 2. Materials and Methods

### 2.1. Study and Sample

The study participants came from an ongoing collaboration between the University of Illinois in Urbana and the Autonomous University of San Luis Potosi in Mexico. The city of San Luis Potosi (SLP), Mexico, is the capital and most populous town of the Mexican state of San Luis Potosi. There are approximately 800,000 people who live in San Luis Potosi, and the city is historically known as a mining community. 

### 2.2. Study Participants

Study procedures incorporated recommendations for research with immigrants and vulnerable families [22] and were approved by the University of Illinois Institutional Review Boards (IRB). All subjects gave their informed consent for inclusion before they participated in the study. The study was conducted in accordance with the Declaration of Helsinki, and the protocol was approved by the Ethics Committee (protocol code: 15503, date: 3 May 2016). Recruitment criteria included participants identifying as a Mexican mother with at least one school-aged child at home. Mothers were recruited through ongoing research projects with the Autonomous University of San Luis Potosi. (The ongoing research projects were unrelated to the current study.) Hispanic project staff recruited 21 mothers in San Luis Potosí, Mexico. Recruitment ended once saturation was reached. 

The 21 participants ranged in age from 28 to 41 years old (m = 34.9). Seventeen mothers were married (85%), and more than half of the participants were full-time stay-at-home mothers (76%). Less than half of the sample (47.6%) completed secundaria (which equates to nine years of education), with 28.6% completing only primaria (up to six years of education). According to the Secretary of Labor and Social Welfare in the Mexican Republic, the average income per day for those living in San Luis Potosí was 326.80 pesos a day, which equates to $17 USD [23] (average area income of $370 USD monthly). Approximately 65% of the sample made less than the average income in San Luis Potosí. See Table 1 for additional participant characteristics. 

### 2.3. Data Collection

Focus groups were chosen as the primary source of data because this technique is often used in cross-cultural research and work with ethnic minorities [24]. There is a limited understanding of cultural factors that impact changes in mealtime routines among Mexican families, and focus groups provide an important way to explore new ideas in a group setting [18].

Participants completed a short self-reported demographic and background survey before the focus group interview started. Questions included age, education, number of household members, and employment. The interview guide included questions about mothers’ mealtime routines and was inspired by prior research with Hispanic samples (e.g., [25,26,27]). Three content areas were included: (a) roles families play in food preparation; (b) family cooking and eating practices in their home as adults and children (e.g., similarities, changes, challenges); and (c) cultural attitudes and beliefs about perceived barriers to healthy eating.

All focus groups were conducted in Spanish. Six culturally competent and bilingual Hispanic interviewers who matched on participant ethnicity and sex conducted the focus groups, taking notes and audiotaping. Sessions took place at local schools or community centers. There were four focus groups conducted with an average of five participants (*n* = 21, focus groups lasted an average of 50 min). All mothers received $10 USD in cash for their participation. A light snack and free child-care were also provided.

### 2.4. Analysis

Each digitally recorded focus group was transcribed verbatim in Spanish, translated to English, and then checked for accuracy by multiple trained research assistants. Three bilingual self-identified Hispanic researchers followed the six-step process by Braun and Clarke when conducting thematic analysis [28]. Researchers used thematic analysis because it can be used to identify patterns across the data in relation to participants’ lived experience, their behaviors, and practices. This type of analysis allows for a way to understand what participants’ think, feel, and do [28] (p. 297).

Researchers familiarized themselves with the data through repeated reading and searching for meanings and patterns. Initial codes were formed based on patterns of meaning identified in the data using the English transcripts. Researchers then re-focused the analysis at the broader level of themes by sorting codes and their extracts into potential themes. Major themes were reviewed by rereading the data and extracts under each category, ensuring fit and sufficient data to support the themes. Themes were then considered according to the data as a whole and defined by determining what aspects of the data captured each theme [28]. Researchers also looked through the Spanish transcription to confirm meaning with the major themes. All three coders performed each of the steps.

Multiple strategies were used to enhance data quality and integrity. Field staff (e.g., recruiters and interviewers) were Hispanic bilingual/bicultural graduate and undergraduate students who underwent extensive training. Each interviewer provided written field notes for each group, yielding additional insights about social and paralinguistic dynamics. Data quality was enhanced through peer debriefing, an activity aimed to provide an external check on the inquiry process by involving peers to explore potential bias and test working hypotheses [29]. Peer debriefing was performed during the coding phase to discuss any discrepancies throughout the process. To ensure methodological integrity, each step of the process was documented using memos to elaborate ideas about the data and help expand descriptive explanations, which also allowed for an audit trail [30]. Members met at each stage of the coding process to discuss the alignment of their coding and to ensure they were capturing forms of diversity in the data, as well as to ground the findings in the evidence [31]. Each team member defended the reasoning behind their codes when discrepancies arose, and then all members came to an agreement through extensive discussion and confirmation with the interview notes. Each member also checked the initial codes of the other team members to ensure reliability among a priori categories. Lastly, data display examples and a process visualization were created to ensure transparency in the analysis process.

## 3. Results

### 3.1. Major Themes

The thematic analysis resulted in three major themes and four corresponding subthemes: 1. Mothers do most of the visible work involving mealtime routines; subtheme 1: children influence the food environment; subtheme 2: fathers can present barriers to healthy eating. 2. Family meals are different today, and globalization is a contributing factor; subtheme 1: technology and electronics are distractions to shared family time; subtheme 2: work barriers and the changing workforce impact family meals. 3. Family mealtimes are shifting to weekend events.

#### 3.1.1. Theme 1: Mothers Do Most of the Visible Work Involving Mealtime Routines 

Mothers discussed how they are typically the main person in the family to cook and shop. Mothers said things like, “During the week no one helps me [shop or cook]. During the week normally I decide, and I know what I’m going to make because they are not there, so there is no way for them to give me a hand”. Mothers contribute a large amount of both time and effort to prepare food and meals for their family; however, mothers acknowledged that they are not the only ones contributing to the decision-making around mealtimes.

##### Subtheme 1.1: Children Influence the Food Environment

Children emerged as a persistent driving force in mealtime patterns and cooking, as mothers shared how children are major influencers of what is being purchased and made for meals at home. Mothers discussed how their children influence their mealtime practices, “It’s very difficult that the majority of kids don’t like vegetables, and for my son it is quite difficult because he doesn’t like it, so we battle too much for them to eat vegetables”. One mother would make vegetables or healthy options but end up being the only one who would eat them. She shared, “I just make the nopales and I end up eating them myself because no one wants them”.

Mothers also talked about how shopping with kids can become burdensome because they ask for certain foods and it can become difficult to deny their requests. Consequently, some mothers feel like they are in constant battles and/or negotiation with their children; for example, “she [my daughter] tells me, ‘I don’t like the meat, mama’. Then I battle a lot with her, but I end up buying her a filet of fish one in a while”.

##### Subtheme 1.2: Fathers Can Present Barriers to Healthy Eating

Mothers discussed how fathers can often present challenges to making healthy food changes within the home. For example, several mothers discussed how their husbands did not want to eat vegetables. One mother said, “my husband is thin. My son and I are the only chubby ones, so he [my husband] doesn’t want veggies”. Another shared, “my husband doesn’t like vegetables either, but I tell him I am not giving you poison so you have to eat it all”.

#### 3.1.2. Theme 2: Family Meals Are Different Today, and Globalization Is a Contributing Factor

Globalization is believed to play a role in the development of unhealthy eating habits. Most of the mothers in this study grew up in an era with a rapidly changing food landscape that they found difficult to navigate now that they are parents. Mothers discussed how food availability, cooking patterns, and preparation are different today compared to the past. They shared how their own mothers would cook with natural products and take time to sit together for family meals when they were growing up. Now, frozen food and preservatives have usurped the more natural products of the past. These changes were especially contrasted with their lives as children when they were growing up. Several mothers discussed how their mothers took the time to make healthier, natural, home-cooked meals. One mother shared, “All the meals with my parents were with vegetables, chicken, soup, and water, but now… the truth is the easiest is soda, and that which is fried, and sometimes I try not to make things like that but they don’t want vegetables”.

Mothers discussed how the foods today are less fresh compared to the past and also how foods that are consumed are less traditional, replaced easily by quick and unhealthy meals. One mother described, “now we have a lot of junk food and it’s what we find in the supermarkets. Before everything was more natural”. Mothers also feel that time constraints make the convenience of readily available foods especially appealing, and as such they often choose the fastest option even if it is unhealthy, as this mother shared, “More than anything it’s the economy because in my time [as a child] they didn’t take us to eat burgers or things like that. Now mothers go for the fastest option. Which is food like burgers, tacos, and all that, but it wasn’t like this before”.

##### Subtheme 2.1: Technology and Electronics Are Distractions to Shared Family Time

Technology was mentioned as commonly present during family mealtimes. Most mothers in the study were not happy about this, discussing how technology use, such as television, phones, or tablets impeded family time during a shared meal. Despite this, it was frequently allowed at the dinner table. Mothers recognized their frustration with technology, even though it tended to be a common practice that families were often reluctant to give up. For instance, “yes, we eat with the T.V. on. I don’t want to lose the telenovela. Well, and the little ones don’t want to miss the cartoons”.

In addition to eating in front of the T.V., families also recognized that the programming they watched could have an influence on their dietary behaviors, as one mother points out, “television [influences my kids’ preferences] because they will see pizza and then they want pizza. They see fries, then they want fries. They don’t promote salad, or water, or juice”.

##### Subtheme 2.2: Work Barriers and the Changing Workforce Impact Family Mealtimes

Some mothers also attributed changes in family mealtime practices to reduced time availability and employment compared to what their own mothers experienced when they were growing up. A large reason for this added barrier was that some mothers in these focus groups were involved in paid work, and they contrasted this with how their mothers never worked outside the home and stayed home as caregivers. One mother shared, “The biggest barrier is lack of money, if I didn’t work it would be another thing, we would have more time to dedicate to my kids, to cook for them more healthily…to eat healthily it costs, for me that is the major problem, the economy”. Conflicting work schedules were also recognized as limiting the frequency of shared mealtimes for many mothers. For instance, “well we eat together on the weekends because on weekdays some of us work. I work different shifts compared to my husband, but on the weekends, we try to eat together”.

#### 3.1.3. Theme 3: Family Mealtimes Are Shifting to Weekend Events 

Several families discussed shifting shared mealtimes to the weekend when there were fewer obligations, and everyone could get together. For example, one mother stated, “well during the week it is different because my husband is not there during the meal hour, neither is he there in the morning so I eat breakfast alone. Though at mealtime my husband isn’t there, my kids are there. Only on the weekends, on Sunday is when we are all there for the three meals together”. Mothers shared that they tended to make an effort to eat together over the weekend, usually with extended family. Eating out was often coupled with eating together as a family on the weekends. Some families however could not afford to eat out due to the cost and large family size, “since we are six, well if I buy them all tacos… it’s just more feasible to make them at home because we are a lot in the family, so I buy them [meals outside the home], but very rarely”.

## 4. Discussion

The purpose of this study was to examine Mexican mothers’ perceptions of mealtime routines and changes in the family food environment. Findings from this study highlight the intersection of culture and gender norm changes, and how this appears to be influencing mealtime routines within the family household. Mothers shared that fathers and children play a larger role in mealtime logistics and the home food environment today, determining the types of foods eaten in the home and often creating barriers to healthy eating. This builds upon other research conducted with Hispanics that have migrated to the United States, in which mothers play a fundamental role in household shopping and cooking (e.g., [31,32,33,34]), as mothers are natural household leaders when it comes to meal planning, cooking, and mealtime routines. However, both fathers and children can counteract this leadership, making unhealthy food requests and challenging the mother’s authority in purchasing and cooking choices [35,36]. The persistence of children in asking for unhealthy food choices, putting snacks or foods in the shopping cart without permission, or refusing to eat certain foods often led to mothers giving in to their children’s preferences, with little to no strategies to combat these unwelcome requests or demands. Some suggested strategies focus on including the child throughout the shopping and meal preparation process, allocating specific tasks to children as a way to get them involved and keep them busy. Several studies have found that when children help with cooking, they are more likely to try the foods they helped prepare, especially vegetables [37,38,39]. Children also learn by example, when partaking in grocery store trips, they often influence what is purchased, and if parents negotiate and teach shopping tactics and model healthy choices, children have the potential to increase healthy food purchasing [33,40]. 

Historically, mothers have served as primary caregivers [41]; however, in recent decades family-based gender roles and household responsibilities have changed [42]. These shifts have been influenced by increasing employment opportunities outside the home for both mothers and fathers [43]. As a result, children’s time with fathers has not only increased, but fathers have also begun to take on more responsibility in organizing their child’s mealtimes and feeding their young children on a weekly basis [44]. Studies of Hispanic fathers suggest that fathers often disagree with mothers about food choices and can act as a barrier to promoting healthy food preferences and mealtime routines (e.g., [45]), which is consistent with this study and a similar one that included Mexican immigrants in the United States [46]. However, it is imperative that research examine fathers’ perceptions of the family food environment as well, as some research suggests that fathers may perceive their role in a different, and perhaps complementarily way to that of mothers (e.g., [35]). Interviewing mothers and fathers together may also yield important insights into how parental perceptions intersect with roles in the mealtime environment. Ultimately, intervention and prevention programs should explore how changes in cultural norms around gender can be used to help empower parents to work together to make healthier eating decisions for the whole family. 

Additional barriers to maintaining healthy mealtime routines and diet included the accessibility of healthy foods and convenience of unhealthy foods, technology as distractions, and work/time conflicts. Mothers described a change in food options, from fresh and local foods to more frozen and instant food options, which has been discussed in previous literature [10] and can be described as a shift away from the traditional Mexican diet. Mothers also discussed this as a shift from their own childhoods, opting for cooking quick and easy foods, which were often fried or instant foods. Unsurprisingly, a recent Mexican study found a highly processed diet to be related to lower diet quality [47]. 

Similarly, a critical component of globalization and changing values is technology use [48]. Having the television on or using phones during meals can preclude family connection in addition to being detrimental to one’s health [44,46,49]. In the current study, mothers discussed how technology was a common accompaniment to family mealtimes and seemed to be an intractable family practice. Many were frustrated with their children’s behaviors around technology. They shared how technology impedes quality family time during meals, often distracting conversation and eating. Positive communication is one possible mechanism that makes shared family meals protective, and devices threaten to weaken this relationship. Device presence at dinner and parents’ ambivalence towards it seems to be a common theme in the literature (e.g., [33]). Perhaps exploring ways in which families can set goals around reducing T.V. time, rather than completely eliminating it, may be helpful in changing behavior, as behavior reduction seems to be more effective in long-term behavior change [46]. Exploring various strategies to help parents to set technology rules that work best for each family may help parents to feel empowered and encourage ongoing conversation around screen time limits during mealtimes [44]. 

In terms of maintaining family meals, participants noted a shift towards weekend mealtimes as a way to spend time together as a whole family and to deal with work and time conflicts throughout the week. These findings parallel other research conducted in the United States examining the mealtime environment with Hispanic families [33,34]. Weekend mealtimes often allowed more family members to be present. However, weekends were also a time when mothers wanted a break from cooking and other household duties, making eating out enticing. Although these times can be characterized as unifying and connecting, eating out is associated with higher total energy intake, less consumption of micronutrients, and a higher consumption of fat [50]. 

Most mealtime studies have focused on Hispanic families that have immigrated to the United States. This paper sheds light on how the challenges families are facing in Mexico mirror those of immigrant Hispanic families in the United States in many ways. Recognizing the similarities across the two countries (especially pertinent because the United States has a large population of Mexican immigrants) may support the adaptation for Mexican audiences of successful health promotion interventions evaluated in the US. For example, the program *Abriendo Caminos,* has been successful in improving Hispanic mothers’ dietary behaviors in the US [49,51]. The current study suggests that these types of programs may be modified for use with Hispanic families in Mexico. Additionally, finding ways to empower mothers as leaders by assisting families in maintaining healthy cultural food traditions, gaining the support of fathers, and strategizing ways to mitigate unhealthy global factors, such as technology use during mealtimes, can be critical components in combatting health disparities and the high obesity prevalence among Hispanic families.

### Limitations

Some limitations should be considered. First, we did not measure remote acculturation or family health variables, such as obesity status. Future studies should assess how remote acculturation influences health and family mealtime routines using validated remote acculturation measures [3,5]. Additionally, examining anthropometric profiles would also allow researchers to explore the interplay between weight status and parental perceptions of the family food environment. Mixed methods approaches may yield important insights into this area in future research. Second, fathers were not included in this study. Current research has pointed to the importance of including fathers’ perspectives in addition to mothers in these types of studies, and in future research we plan to work with fathers directly to understand their perceptions and experiences of mealtimes and family routines. Despite the limitations suggested, this study brings unique elements that bridge a number of gaps in the literature. First, this study examines Mexicans living in Mexico. These data provide insight into how mothers’ perspectives on modernization play a role in family dynamics and mealtime routines. Exploring behavior changes in mealtime and family practices from a cultural perspective is critical, and the perceptions of the Mexican mothers in this study provided an understanding of how family mealtime routines are experienced. Cultural shifts with regard to gender roles, child behaviors, and the food environment, are likely to play a powerful role in shaping health outcomes including child and adult obesity among Hispanic families. Culturally-tailored family health programs may have the potential to help families to adapt to these changes in positive and enduring ways.

## Figures and Tables

**Table 1 children-09-01045-t001:** Demographic Characteristics of Focus Group Participants (*n* = 21).

	San Luis Potosí, MX (*n* = 21)
Characteristic (%)	
Mother’s Age (in years; *M* ± *SD*)	34.9 ± 6.4
Education	
Primaria (Elementary equivalence)	6 (28.6)
Secundaria (Middle school equivalence)	10 (47.6)
Preparatora (High school equivalence)	2 (9.5)
More than Preparatora	3 (14.3)
Number of household members (*M* ± *SD*)	4.6 ± 1.6
Relationship Status	
Married/living with partner	17 (85)
Single	2 (7.5)
Divorced/Widowed	2 (7.5)
Employment Status	
Employed outside the home	5 (24.0)
Stay at home mother	16 (76.0)

## Data Availability

Not applicable.

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
