# Peer review of "Cultural Influences on Family Mealtime Routines in Mexico: Focus Group Study with Mexican Mothers"

_children, 2022, doi:10.3390/children9071045_

Round 1

Reviewer 1 Report

Review notes for “Cultural Influences on Family Mealtime Routines in Mexico: A Study with Mexican Mothers”
General comments
- This study explored the perception of Mexican mothers on cultural influences of mealtime routines. The study adds to the limited literature on the challenges faced by nonimmigrant Mexican population on the area of maintaining healthy mealtime routines and provides some foundation further research needed to inform programs and policies. Other than some minor methodological deficits, the paper is written and organized well.
Specific comments
Title:
- Does not provide a clue on the type/design of the study. It is helpful to modify the title, so it does reflect the nature of the study
Abstract:
- The abstract is concise. However, the suggested “three main themes” should be stated clearly.
- Also, the nature of the study, though implied, not clearly stated in the abstract...
- The meaning of the recommendation would be enhanced if authors include some examples of the mechanisms/strategies of empowering the mothers in their recommendation
Introduction:
- Provides a concise summary of the background to the problem with relevant literature and makes a case for why the current study
- The purpose of the study is better summarized as a paragraph at the end of the introduction section. The way it is presented currently reads more like a thesis/dissertation than a manuscript.
Methods and materials:
- This section provides the data collection method/technique (i.e., focus group discussions) and the analysis (i.e., thematic) but misses to describe the overall methodology and theoretical framework that underpins the methodology. E.g., what is the specific qualitative methodological approached used in the study (phenomenology, ethnography, narrative research, case study, etc.) and why?
- Other sections of the methodology are well described.
Results:
- Well organized by main themes and subthemes with direct quotes supporting each main theme and sub-theme.
- Lines 225-227: “Mothers also feel that financial constraints are less of a burden and often choose the fastest option even if it is un-healthy, as this mother shared:” this sentence is not clear. Are financial constraints burdensome or not burdensome to mothers?
-
Discussions & Limitations:
- Key results are adequately discussed with supporting literature.
- Perhaps authors could have included as part of the background characteristics the
anthropometric profile of the participants and their families to provide more context to the
issues raised by research participants…Did they have overweight/obese children? Were they
themselves overweight or obese? Do we know whether these characteristics modify the
perceptions/opinion gathered from the participants. How were each participant selected? etc. If
these aspects were not addressed, authors may want to indicate this in the limitation section.
- As part of the recommendation, please give specific examples of what empowers mothers.
Ensure your recommendation in the abstract section and in the main text aligns well.

Author Response

Author responses to reviewer comments are in blue.

Review notes for “Cultural Influences on Family Mealtime Routines in Mexico: A Study with Mexican Mothers”
General comments
- This study explored the perception of Mexican mothers on cultural influences of mealtime routines. The study adds to the limited literature on the challenges faced by nonimmigrant Mexican population on the area of maintaining healthy mealtime routines and provides some foundation further research needed to inform programs and policies. Other than some minor methodological deficits, the paper is written and organized well.
Specific comments
Title:
- Does not provide a clue on the type/design of the study. It is helpful to modify the title, so it does reflect the nature of the study

Thank you for this insight. We have changed the title to include that it was a Focus Group study with Mexican Mothers.

Abstract:
- The abstract is concise. However, the suggested “three main themes” should be stated clearly.
- Also, the nature of the study, though implied, not clearly stated in the abstract...
- The meaning of the recommendation would be enhanced if authors include some examples of the mechanisms/strategies of empowering the mothers in their recommendation

We have clarified the main themes, included the nature of the study, and included some of the recommendations in the abstract.

Introduction:
- Provides a concise summary of the background to the problem with relevant literature and makes a case for why the current study
- The purpose of the study is better summarized as a paragraph at the end of the introduction section. The way it is presented currently reads more like a thesis/dissertation than a manuscript.

In response to these comments, we have written a summary paragraph to introduce the main purpose of the study.

Methods and materials:
- This section provides the data collection method/technique (i.e., focus group discussions) and the analysis (i.e., thematic) but misses to describe the overall methodology and theoretical framework that underpins the methodology. E.g., what is the specific qualitative methodological approached used in the study (phenomenology, ethnography, narrative research, case study, etc.) and why?
- Other sections of the methodology are well described.

Thank you, we have included a rationale and an explanation of our method in the analysis section.

Results:
- Well organized by main themes and subthemes with direct quotes supporting each main theme and sub-theme.
- Lines 225-227: “Mothers also feel that financial constraints are less of a burden and often choose the fastest option even if it is un-healthy, as this mother shared:” this sentence is not clear. Are financial constraints burdensome or not burdensome to mothers?

Thank you for catching this, we have clarified that sentence.

Discussions & Limitations:
- Key results are adequately discussed with supporting literature.
- Perhaps authors could have included as part of the background characteristics the anthropometric profile of the participants and their families to provide more context to the issues raised by research participants…Did they have overweight/obese children? Were they themselves overweight or obese? Do we know whether these characteristics modify the perceptions/opinion gathered from the participants. How were each participant selected? etc. If these aspects were not addressed, authors may want to indicate this in the limitation section.
- As part of the recommendation, please give specific examples of what empowers mothers.
Ensure your recommendation in the abstract section and in the main text aligns well.

Unfortunately, we did not collect anthropometric data with these participants but have included this as a limitation. We agree that would have added to the study. We have also included clear recommendations throughout the discussion section.

Reviewer 2 Report

This study doesn' have a good statistic analysis. 

There isn't the questionnaire to which the women responded. 

There are many English language errors. 

I think that probably authors should rewrite the work using an appropriate statistic analysis to discuss properly their results.

Author Response

Author responses to reviewer comments are in blue.

This study doesn' have a good statistic analysis. 

There isn't the questionnaire to which the women responded. 

There are many English language errors. 

I think that probably authors should rewrite the work using an appropriate statistic analysis to discuss properly their results.

Thank you for your comments. Since this was a qualitative study, we did not conduct statistical analysis. The participants did not answer a survey about the research questions, only focus group/open-ended questions were used. While we did collect basic demographic information, this was only used to present sample characteristics as background information for readers in Table 1. We have described the qualitative methods in the methodology section. We have also done another review of the English language errors.

Reviewer 3 Report

This is a highly interesting manuscript reporting Mexican mother’s perceptions of factors affecting healthy eating. The qualitative analysis has been conducted and reported thoroughly, and the manuscript is well written. However, some questions (mostly minor) arose which needs to be addressed:

- In the abstract (line 20) a phrase “dietary pattern” is used even though the study does not describe dietary patterns as they are generally referred in the literature (descriptions of overall diet through for example principal components analysis or dietary indices). Maybe it could be replaced with just a word “diet”. Frequently used expression “mealtime routines” is also a bit unclear and I think it should be defined in the manuscript, what authors mean by it.

- In the abstract it is reported that 85% were stay-at-home mothers, but in the Methods section the respective percentage was 76%. Which one is correct?

- In the abstract (line 23) one major theme is reported as “changes in when mealtime typically takes place”. It would be more exact to say it was about shared family meals shifting towards weekends. Based on the abstract only, I thought that usual timing of meals within a day was shifting.

- Isn’t the abbreviation of the Organization of Economic Co-operation and Development is OECD rather than OCED?

- As a reader very unfamiliar with Hispanic culture, I was intrigued by the emphasis on the role of mothers in matters related to food and cooking as described in lines 55-59. The results of the present manuscript indicate that mothers still do the most work but fathers and children have more influence than traditionally, and their influence is towards unhealthier direction. In main conclusions, the authors conclude that a solution might be empowering mothers as leaders. In the discussion (lines 305-307) authors nicely consider that changing gender roles may be used to help parents work together to make healthier decisions. In my mind, this point of view could be emphasized in the conclusions also. However, this is just an opinion, not even a suggestion for change.

- I’m also wondering, could it also be that mothers tend to use children and fathers as a socially desirable excuse for unhealthy habits? That is of course something that the present data does not tell, but the notion the authors make in the discussion (lines 300-304) about interviewing fathers also is very important. Maybe this is something that could be mentioned also in the limitations section?

- It is not clear how the authors end up with the sample size of 21 (because of saturation, practical reasons etc?), neither is it discussed whether the sample size affects the conclusions. Additionally, participants were recruited from ongoing research projects. May this have influenced the results for example through selection bias (maybe not an issue in a qualitative study though)?

- Table 1, Education and Marital status rows could benefit from captions (similar to Employment status), line “None” is not needed for education, because there weren’t any mothers with reporting “none” education.

- It is mentioned (lines 138-139) that transcribed interviews were translated and the process is described nicely, but the thematic analysis was done by bilingual researchers. Was the analysis done using original or translated version?

- In line 151 it is stated that “data collection was led by a European American faculty member”, but for me it’s relevance is not clear. I’m not familiar with the faculty in question.

- How many interviewers there were altogether? Did I understand correctly that all interviewers participated all interviews? If not, do you think this has any effect on the results?

- Does “peer debriefing” described in lines 155-158 consider data collection (interviews) or data analysis phase?

- Did you use any special software to facilitate the analysis or was it done by hand?

- Numbering of the themes and subthemes is confusing as it is written in the result section (lines 172-277). Is it possible to number major themes 1., 2. and 3. and subthemes 1.1, 1.2, and 2.1, 2.2 etc. Is the Table 2 missing from the submitted manuscript? I feel it may have improved understanding, but I failed to find it.

- A quote in lines 199-200 is interesting, because it describes child preferring fish over meat. In general, fish is healthier option than meat. Children’s influence on food environment is presented in the manuscript mainly as detrimental, but this example describes the opposite. Was this the only example, or was there more evidence about the positive impact of children? I’m thinking this, because younger ones are sometimes more open to adopt (culturally) new behaviors, also the healthy ones.

- Technology use is presented in the present manuscript as a barrier for healthy eating, but not many reasons for that are presented. In my mind, technology is linked with the increased rates of obesity mainly through decreased physical activity. The link between unhealthy eating could be justified better than just “as a form of distraction”. In addition, in lines 318-320 it is stated that technology is a distraction and detrimental to one’s health, but I’m wondering, whether the reference nro 49 is correct in this case. It describes results from an intervention, but based on a quick look there is nothing about technology.

- There is something else wrong in the reference list also. In line 346 there are references numbered 51 and 52, but the reference list ends to a number 50.

- Final conclusions in the manuscript (lines 347-351) are accompanied with a reference (nro 32). Hence, it is unclear, whether these conclusions are made based on the results of the present study or based on the reference.

Author Response

Responses by the author to reviewer comments are in blue. 

This is a highly interesting manuscript reporting Mexican mother’s perceptions of factors affecting healthy eating. The qualitative analysis has been conducted and reported thoroughly, and the manuscript is well written. However, some questions (mostly minor) arose which needs to be addressed:

- In the abstract (line 20) a phrase “dietary pattern” is used even though the study does not describe dietary patterns as they are generally referred in the literature (descriptions of overall diet through for example principal components analysis or dietary indices). Maybe it could be replaced with just a word “diet”. Frequently used expression “mealtime routines” is also a bit unclear and I think it should be defined in the manuscript, what authors mean by it.

Yes, this makes sense, we have included the definition for mealtime routine and changed the phrase of dietary pattern to diet.

- In the abstract it is reported that 85% were stay-at-home mothers, but in the Methods section the respective percentage was 76%. Which one is correct?

Thank you for pointing out this error. The correct percent is 76% and we have updated that in the abstract.

- In the abstract (line 23) one major theme is reported as “changes in when mealtime typically takes place”. It would be more exact to say it was about shared family meals shifting towards weekends. Based on the abstract only, I thought that usual timing of meals within a day was shifting.

We have clarified this section.

- Isn’t the abbreviation of the Organization of Economic Co-operation and Development is OECD rather than OCED?

Yes, thank you for catching this error. We have changed this throughout the manuscript.

- As a reader very unfamiliar with Hispanic culture, I was intrigued by the emphasis on the role of mothers in matters related to food and cooking as described in lines 55-59. The results of the present manuscript indicate that mothers still do the most work but fathers and children have more influence than traditionally, and their influence is towards unhealthier direction. In main conclusions, the authors conclude that a solution might be empowering mothers as leaders. In the discussion (lines 305-307) authors nicely consider that changing gender roles may be used to help parents work together to make healthier decisions. In my mind, this point of view could be emphasized in the conclusions also. However, this is just an opinion, not even a suggestion for change.

Thank you for noting your point of view. We have further emphasized this point throughout the discussion.

- I’m also wondering, could it also be that mothers tend to use children and fathers as a socially desirable excuse for unhealthy habits? That is of course something that the present data does not tell, but the notion the authors make in the discussion (lines 300-304) about interviewing fathers also is very important. Maybe this is something that could be mentioned also in the limitations section?

We did include that future studies should interview fathers and included that in our limitations section. We agree that that could be factor and have seen in other studies that mothers can feel guilty and will give into their children’s desire or want to eat with their spouse for a second time to please them because they arrived after the mother and the children ate.

- It is not clear how the authors end up with the sample size of 21 (because of saturation, practical reasons etc?), neither is it discussed whether the sample size affects the conclusions. Additionally, participants were recruited from ongoing research projects. May this have influenced the results for example through selection bias (maybe not an issue in a qualitative study though)?

We have included more details about the sample size and recruitment processes.

- Table 1, Education and Marital status rows could benefit from captions (similar to Employment status), line “None” is not needed for education, because there weren’t any mothers with reporting “none” education.

Thank you for noting this. We had subheadings before but they must have been unintentionally deleted in the process of editing.

- It is mentioned (lines 138-139) that transcribed interviews were translated and the process is described nicely, but the thematic analysis was done by bilingual researchers. Was the analysis done using original or translated version?

We have clarified the procedure we took in the analysis section. We coded the English transcripts but used the Spanish transcription to confirm meaning.

- In line 151 it is stated that “data collection was led by a European American faculty member”, but for me it’s relevance is not clear. I’m not familiar with the faculty in question.

We originally included this to clarify each person’s role when conducting research with another country that speaks another language but have deleted this sentence since it is not relevant to the current study.

- How many interviewers there were altogether? Did I understand correctly that all interviewers participated all interviews? If not, do you think this has any effect on the results?

We had the same two interviewers in each of the focus groups. Then those two same researchers plus one other researcher coded and worked with the data to understand themes.

- Does “peer debriefing” described in lines 155-158 consider data collection (interviews) or data analysis phase?

We conducted peer debriefing during the data analysis phase. We have clarified this in the manuscript.

- Did you use any special software to facilitate the analysis or was it done by hand?

Coding was done using word documents and by hand.

- Numbering of the themes and subthemes is confusing as it is written in the result section (lines 172-277). Is it possible to number major themes 1., 2. and 3. and subthemes 1.1, 1.2, and 2.1, 2.2 etc. Is the Table 2 missing from the submitted manuscript? I feel it may have improved understanding, but I failed to find it.

Yes, I agree that the labeling process makes more sense, we have changed that. We also did not end up including table 2 because it contained the same quotes as the manuscript. We have now taken out that phrase.

- A quote in lines 199-200 is interesting, because it describes child preferring fish over meat. In general, fish is healthier option than meat. Children’s influence on food environment is presented in the manuscript mainly as detrimental, but this example describes the opposite. Was this the only example, or was there more evidence about the positive impact of children? I’m thinking this, because younger ones are sometimes more open to adopt (culturally) new behaviors, also the healthy ones.

We have also noticed this to be the case in other studies where children of Hispanic immigrant mothers often prefer healthy foods but mothers do not believe that to be true. Although we have seen this in other studies we did not see that as the norm across the focus group here. Mexican children do seem to have slight diet differences compared to U.S. children but there have not been many studies on this.

- Technology use is presented in the present manuscript as a barrier for healthy eating, but not many reasons for that are presented. In my mind, technology is linked with the increased rates of obesity mainly through decreased physical activity. The link between unhealthy eating could be justified better than just “as a form of distraction”. In addition, in lines 318-320 it is stated that technology is a distraction and detrimental to one’s health, but I’m wondering, whether the reference nro 49 is correct in this case. It describes results from an intervention, but based on a quick look there is nothing about technology.

This might have been an error, we have included a few more citations that speak to screentime and the effects on health. #49 speaks to an intervention that included a technology unit to support diet with Hispanic families.

We added ‘to shared family time’ to the subtheme to explain ‘the distraction to what’ open question and clarified it further in the discussion section.

- There is something else wrong in the reference list also. In line 346 there are references numbered 51 and 52, but the reference list ends to a number 50.

We have corrected this error.

- Final conclusions in the manuscript (lines 347-351) are accompanied with a reference (nro 32). Hence, it is unclear, whether these conclusions are made based on the results of the present study or based on the reference.

We see how that can be confusing. That final conclusion is meant to stand alone as these are the findings for this study not based on a different study although they are related but with different populations.